# Deterministic Assembly Processes Strengthen the Effects of β-Diversity on Community Biomass of Marine Bacterioplankton

Feng-Hsun Chang,[a] Jinny Wu Yang,[b] Ariana Chih-Hsien Liu,[a] Hsiao-Pei Lu,[c] Gwo-Ching Gong,[d,e] Fuh-Kwo Shiah,[a,d,f] Chih-hao Hsieh[a,f,g,h,i]

aInstitute of Oceanography, National Taiwan University, Taipei, Taiwan
bEcology and Evolutionary Biology, University of Michigan, Ann Arbor, Michigan, USA
cDepartment of Biotechnology and Bioindustry Sciences, National Cheng Kung University, Tainan, Taiwan
dInstitute of Marine Environment and Ecology, National Taiwan Ocean University, Keelung, Taiwan
eCenter of Excellence for the Oceans, National Taiwan Ocean University, Keelung, Taiwan
fResearch Center for Environmental Changes, Academia Sinica, Taipei, Taiwan
gInstitute of Ecology and Evolutionary Biology, National Taiwan University, Taipei, Taiwan
hDepartment of Life Science, National Taiwan University, Taipei, Taiwan
iNational Center for Theoretical Sciences, Taipei, Taiwan

**ABSTRACT** The presence of more species in the community of a sampling site ($\alpha$ diversity) typically increases ecosystem functions via nonrandom processes like resource partitioning. When considering multiple communities, we hypothesize that higher compositional difference ($\beta$ diversity) increases overall functions of these communities. Further, we hypothesize that the $\beta$ diversity effect is more positive when $\beta$ diversity is increased by nonrandom assembly processes. To test these hypotheses, we collected bacterioplankton along a transect of 6 sampling sites in the southern East China Sea in 14 cruises. For any pairs of the 6 sites within a cruise, we calculated the Bray-Curtis index to represent $\beta$ diversity and summed bacterial biomass as a proxy to indicate the overall function of the two communities. We then calculated deviation of observed mean pairwise phylogenetic similarities among species in two communities from random to represent the influences of nonrandom processes. The bacterial $\beta$ diversity was found to positively affect the summed bacterial biomass; however, the effect varied among cruises. Cross-cruise comparison indicated that the $\beta$ diversity effect increased with the nonrandom processes selecting for phylogenetically dissimilar species. This study extends biodiversity-ecosystem functioning research to the scale of multiple sites and enriches the framework by considering community assembly processes.

**IMPORTANCE** The implications of our analyses are twofold. First, we emphasize the importance of studying $\beta$ diversity. We expanded the current biodiversity-ecosystem functioning framework from single to multiple sampling sites and investigated the influences of species compositional differences among sites on the overall functioning of these sites. Since natural ecological communities never exist alone, our analyses allow us to more holistically perceive the role of biodiversity in natural ecosystems. Second, we took community assembly processes into account to attain a more mechanistic understanding of the impacts of biodiversity on ecosystem functioning.

**KEYWORDS** $\beta$ diversity, biodiversity-ecosystem functioning, community assembly processes, East China Sea, homogeneous versus heterogeneous selection

Address correspondence to Feng-Hsun Chang, fhchang422@ntu.edu.tw, or Chih-hao Hsieh, chsieh@ntu.edu.tw.

The authors declare no conflict of interest.

Decades of studies have shown that having more species in a single ecological community of a sampling site, i.e., having higher $\alpha$ diversity, often leads to higher ecosystem functions of this community, such as resource use efficiency and community biomass (1, 2). However, when considering multiple communities, the association between the species

compositional difference, i.e., $\beta$ diversity, and the overall functions of these communities is less clear (3). Some studies show a positive association between $\beta$ diversity and the overall functioning of multiple communities (4), but others show that $\beta$ diversity is not important in predicting ecosystem functions (5); two recent studies even suggest that $\beta$ diversity negatively affects the overall functions of multiple communities (6, 7). The differences among ecosystem types might contribute to the variation of the $\beta$ diversity effect. For example, dispersal limitation determines the $\beta$ diversity and the overall function of multiple communities (8), and dispersal limitation is widely acknowledged to be effective at different spatial and temporal scales depending on ecosystem type (9). To mechanistically investigate the relationship between $\beta$ diversity and overall functioning, we chose to depict the community assembly processes that determine the species composition, diversity and subsequently ecosystem functioning of communities. The community assembly processes are worth considering to complement the biodiversity-ecosystem functioning research framework (3, 10).

In fact, community assembly processes are embedded in the calculation of $\beta$ diversity, because $\beta$ diversity quantifies how species compositions differ among multiple communities (11, 12). For example, some studies have statistically partitioned $\beta$ diversity in order to understand whether species composition varies with certain environmental gradients (13, 14). Others have analyzed $\beta$ diversity by implementing multivariate statistical methods (15, 16), neutral-theory-based process models (17, 18), or statistical null model approaches (19–21) in order to infer how the species compositions are determined. When investigating the relationship between $\beta$ diversity and overall functions of multiple communities, the influences of assembly processes are inherently taken into account.

The influence of assembly processes on species compositions and $\beta$ diversity among multiple communities can range from completely stochastic to completely deterministic (22, 23). When the influence is completely stochastic, species' birth, death, presence/absence, and thus population size are all governed by random chance (18, 24). Either due to random environmental fluctuations (i.e., environmental stochasticity) or independent of environments (i.e., demographic stochasticity), species' random birth, death, and dispersal can cause the species compositions to be different (25). At the other extreme, completely deterministic processes reflect species' nonrandom growth, death, and dispersal rates, which possibly result from some biotic (e.g., species competition) or abiotic (e.g., habitat filtering) interactions among species (20, 26, but see 27). Depicting the influence of assembly processes on the stochastic-deterministic spectrum helps explain and predict how diversity changes at different spatial and temporal scales (28–30).

More importantly, depicting the influence of assembly processes may help explain whether $\beta$ diversity enhances, decreases, or exerts no clear effects on the overall functions of multiple communities (8, 31). For example, some species can persist and contribute to $\beta$ diversity due to stochastic processes like random dispersal, but these species are not able to utilize resources and perform ecological functions (24, 32); under such circumstances, $\beta$ diversity does not necessarily enhance the overall functions of those communities. In contrast, the influence of deterministic assembly processes implies selection of species that can capture resources and survive, so that $\beta$ diversity should positively affect the overall functions. We therefore argue that when communities are influenced by assembly processes that are more deterministic and less stochastic, $\beta$ diversity of these communities would have stronger effects on the overall functions of multiple communities.

To depict the assembly processes of multiple communities on the stochastic-deterministic spectrum, we examined the degree of relatedness, i.e., phylogenetic similarity, among species (33, 34), because phylogenetic similarity is considered an imprint left by evolutionary and ecological processes (35, 36). We then apply the null model approach on the phylogenetic similarity among species in a set of multiple communities to infer the relative importance of stochastic versus deterministic assembly processes (20). The null model approach is elaborated in Materials and Methods. In short, the null distribution of phylogenetic similarity is derived by randomization to represent the scenario governed by completely stochastic processes. When phylogenetically

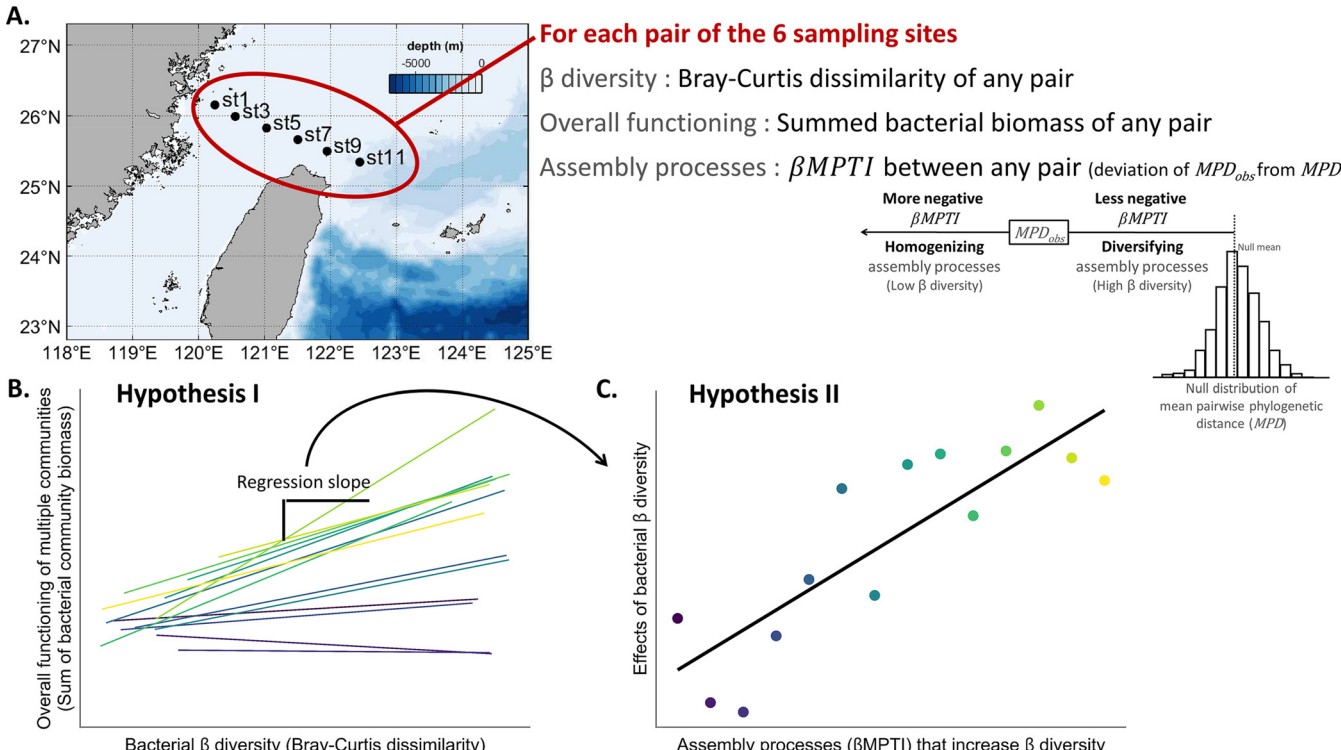

**FIG 1** Schematic plots visualizing the sampling sites, analytical flow, and hypotheses. (A) The six sampling sites in the southern East China Sea and indices of $\beta$ diversity, overall functioning (using biomass as a proxy), and assembly processes of the multiple communities. (Inset) Interpretation of the index of assembly processes ($\beta$MPTI). (B) Expectation of hypothesis I (that bacterial $\beta$ diversity increases the sum of bacterial community biomass). Each colored line represents the linear model that regresses the summed bacterial community biomass on bacterial $\beta$ diversity for each cruise, and the regression slope is an indicator of $\beta$ diversity effect. The regression slopes are shown as colored dots in panel C. (C) Expectation of hypothesis II (that the effect of bacterial $\beta$ diversity on the sum of bacterial community biomass is stronger when $\beta$ diversity is increased more by deterministic and less by stochastic assembly processes).

similar species are selected by homogenizing deterministic processes, the observed phylogenetic similarity will negatively deviate from random. In contrast, when diversifying deterministic assembly processes select for phylogenetically dissimilar species and increase $\beta$ diversity, the observed phylogenetic similarity is expected to positively deviate from random.

To investigate the influences of community assembly processes on the association between bacterial $\beta$ diversity and the overall functions of multiple communities, we collected bacterial samples along a transect of 6 sampling sites in the southern East China Sea (ECS) in 14 cruises (see the map in Fig. 1A). The southern ECS is a hydrographically complex region (37, 38). The inner shelf is dominated by river inputs and coastal currents that are typically nonsaline and rich in terrestrial nutrients (39, 40). In the middle shelf, the Taiwan Warm Current (Taiwan Strait Current) is warm and nutrient depleted at the surface but saline and nutrient rich in the bottom layer (39, 40). To the northeast of Taiwan at the edge of continental shelf, the upwelled subsurface water from the Kuroshio Current brings ample amounts of saline and nutrient-rich water to the ECS (41). Interactions among these water masses vary spatially and seasonally to determine the dynamics of nutrients, chlorophyll $a$, primary production (42), and bacterial biomass and production (43, 44). These spatial and temporal complexities also create among-cruise variations in deterministic versus stochastic assembly processes for bacterial communities in the southern ECS (45).

In this study, we first tested hypothesis I: bacterial $\beta$ diversity increases the summed bacterial community biomass (Fig. 1B). Specifically, for any pair of the 6 sampling sites within a cruise, we calculated the Bray-Curtis index as the bacterial $\beta$ diversity and the sum of bacterial community biomass as a proxy to indicate the overall function of the two sites. Then, the effect of $\beta$ diversity could thus be estimated as the slope when regressing the sum of bacterial biomass against the $\beta$ diversity for each pair of the

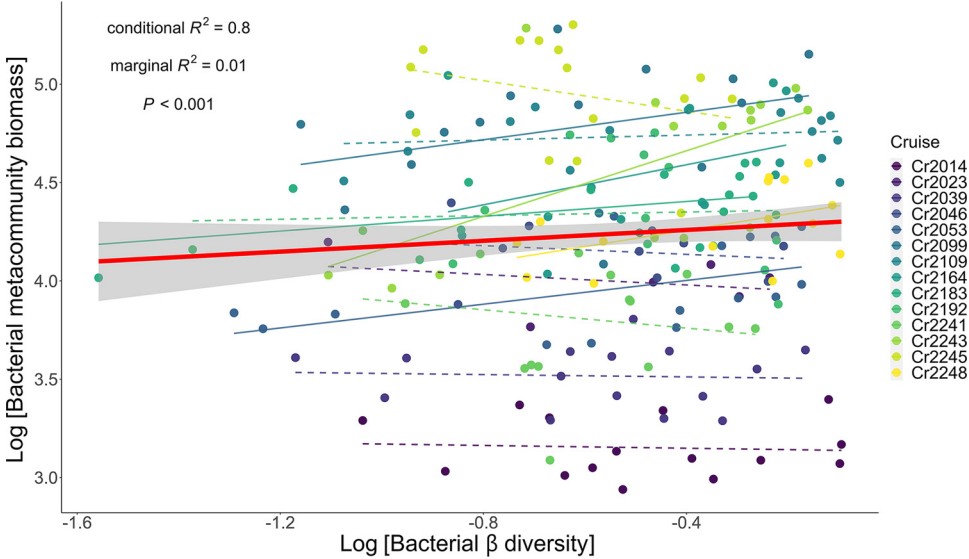

**FIG 2** Association between the summed bacterial community biomass and $\beta$ diversity. The bold red solid line indicates the regression line fitted by the GLMM, with cruise as the random effect. The shaded area represents the confidence interval of regression. The colored dots represent measurements from different sampling cruises. The thin colored lines are the regression lines fitted by a GLM regressing the sum of pairwise bacterial community biomass on $\beta$ diversity for each cruise. These regression lines vary substantially among cruises. Colored solid and dashed lines represent significant and nonsignificant regression slopes, respectively.

6 sampling sites within a cruise (Fig. 1B). Second, we tested hypothesis II: based on cross-cruise comparison, the bacterial $\beta$ diversity effect is stronger (more positive) when $\beta$ diversity is increased more by deterministic and less by stochastic assembly processes (Fig. 1C). That is, we expect that the effects of bacterial $\beta$ diversity (defined as the estimated slope, as explained in Fig. 1B) would be more positive when the observed phylogenetic similarity more positively deviates from random.

## RESULTS

In the southern ECS, physical and chemical environments significantly changed from the inner to outer shelf. At the most inner shelf station, temperature and salinity were significantly lower than at other stations, especially in the springtime ($P < 0.01$) (see Fig. S1 in the supplemental material). This signaled the runoff of the Min River, which significantly increased nitrogen and phosphorus concentrations and caused the chlorophyll *a* concentration to be significantly higher at the most inner shelf station than other stations ($P < 0.01$) (Fig. S1), except for the station northeast of Taiwan (station 9) (Fig. S1). The station at the northeast of Taiwan is impacted by the year-round upwelling of subsurface Kuroshio waters, so that it is generally nutrient rich (37). However, the nutrients supplied by upwelling were lower than those in the river runoff, so that bacterial community biomass at this station was still significantly lower than that at the inner shelf stations (stations 1 and 3; $P < 0.01$) (Fig. S1).

In association with the environmental variations, the bacterial community also showed a clear compositional shift from the inner to outer shelf of the southern ECS (Fig. S2). Specifically, the bacterial communities at the inner shelf were more positively associated with temperature and salinity, while the communities at the outer shelf were more positively associated with total inorganic nitrogen and phosphorus (Fig. S2). These results suggest that the environment is indeed spatially heterogeneous, and such spatial heterogeneity in turn affects the community compositions and biomasses of bacterioplankton in the southern ECS.

For hypothesis I (Fig. 1B), we found that the sum of bacterial community biomass was positively associated with bacterial $\beta$ diversity (univariate regression coefficient = 0.17) (Fig. 2), even after accounting for potential confounding environmental variables (Table 1).

**TABLE 1** Results of GLMM testing of hypothesis I[a]

| Independent variable[b] | Regression coefficient | SE | P |
|---|---|---|---|
| Log (bacterial $\beta$ diversity) | 0.09 | 0.04 | 0.04 |
| Log (salinity) | −3.86 | 0.44 | <0.01 |
| Log (total inorganic nitrogen) | −0.02 | 0.01 | <0.01 |

[a]Hypothesis I is that bacterial Bray-Curtis dissimilarity index increases the sum of bacterial community biomass (Fig. 1A). The model presented here shows the results after backward stepwise selection to statistically control for potential confounding factors. Steps of the backward stepwise selection are listed in Tables S1 and S2. The results indicate that, after accounting for environmental variables, bacterial $\beta$ diversity positively affected the sum of bacterial community biomass. This conclusion is qualitatively the same as that drawn from the univariate GLMM (Fig. 2).

[b]Summed bacterial community biomass was the dependent variable.

Our results thus support hypothesis I to some extent. However, this association was highly variable, judging from the low marginal $R^2$ (P value = 0.01) of the generalized linear mixed-effects model (GLMM). When fitting the generalized linear model (GLM) for each cruise, 8 of 14 regression slopes were nonsignificant (colored dashed lines in Fig. 2), indicating that the effect of $\beta$ diversity was not always positive. In addition, we found that the $\beta$ mean pairwise taxonomic index ($\beta$MPTI) varied significantly among the 14 sampling cruises and seemed to be lower in those 8 cruises (Fig. S3). These results suggest that the effect of bacterial $\beta$ diversity on the sum of bacterial biomass is generally positive, but it could vary with the community assembly processes.

We then tested hypothesis II: based on cross-cruise comparison, the bacterial $\beta$ diversity effect is stronger (more positive) when $\beta$ diversity is increased more by deterministic and less by stochastic assembly processes (Fig. 1C). We first noticed that $\beta$MPTI was generally negative in our study (mean = −1.22; standard deviation = 0.76), indicating that the bacterial communities in the southern ECS were generally subject to homogenizing assembly processes. However, when $\beta$MPTI was less negative, bacterial $\beta$ diversity became higher (univariate regression slope = 0.09 and P value < 0.01) (Fig. 3). These results suggest that $\beta$ diversity of multiple bacterial communities is increased when deterministic assembly processes weaken the influences of homogenization by selecting for phylogenetically dissimilar species.

Through cross-cruise comparison, we further found that when $\beta$MPTI became less negative, the effects of bacterial $\beta$ diversity on the summed bacterial biomass of the two communities became more positive (Fig. 4). This positive relationship remained statistically significant after controlling for bacterial $\alpha$ diversity and environmental factors (Table 2 and Fig. S6). Given these results, hypothesis II is supported, indicating that when nonrandom/deterministic processes select for phylogenetically dissimilar species in the two communities (less negative $\beta$MPTI), $\beta$ diversity is increased and the effect of $\beta$ diversity on the sum of bacterial communities becomes more positive.

## DISCUSSION

In the southern ECS, we first confirmed that the environment is spatially heterogeneous, and the bacterial composition is thereby affected by spatial heterogeneity (Fig. S1 and S2). Moreover, the degree of spatial heterogeneity varied among cruises. These results are in agreement with previous studies in this region (37, 38, 45). Given the heterogeneous environments and bacterial composition, we investigate the association between bacterial $\beta$ diversity and the overall function (using biomass as a proxy) of multiple sampling sites.

In general, we found a positive, but highly variable, effect of bacterial $\beta$ diversity on the summed community biomass of any pair of the 6 sampling sites within a cruise (Fig. 2); however, the $\beta$ diversity effects were not significant in 8 of the 14 cruises (colored dashed lines in Fig. 2). Current research on the relationship between $\beta$ diversity and the overall functions of multiple sites has also found similar patterns, showing that $\beta$ diversity generally has positive effects on ecosystem functions (4, 46–49) but that nonsignificant or even negative $\beta$ diversity effects are not uncommon (6, 7, 50). We acknowledge that

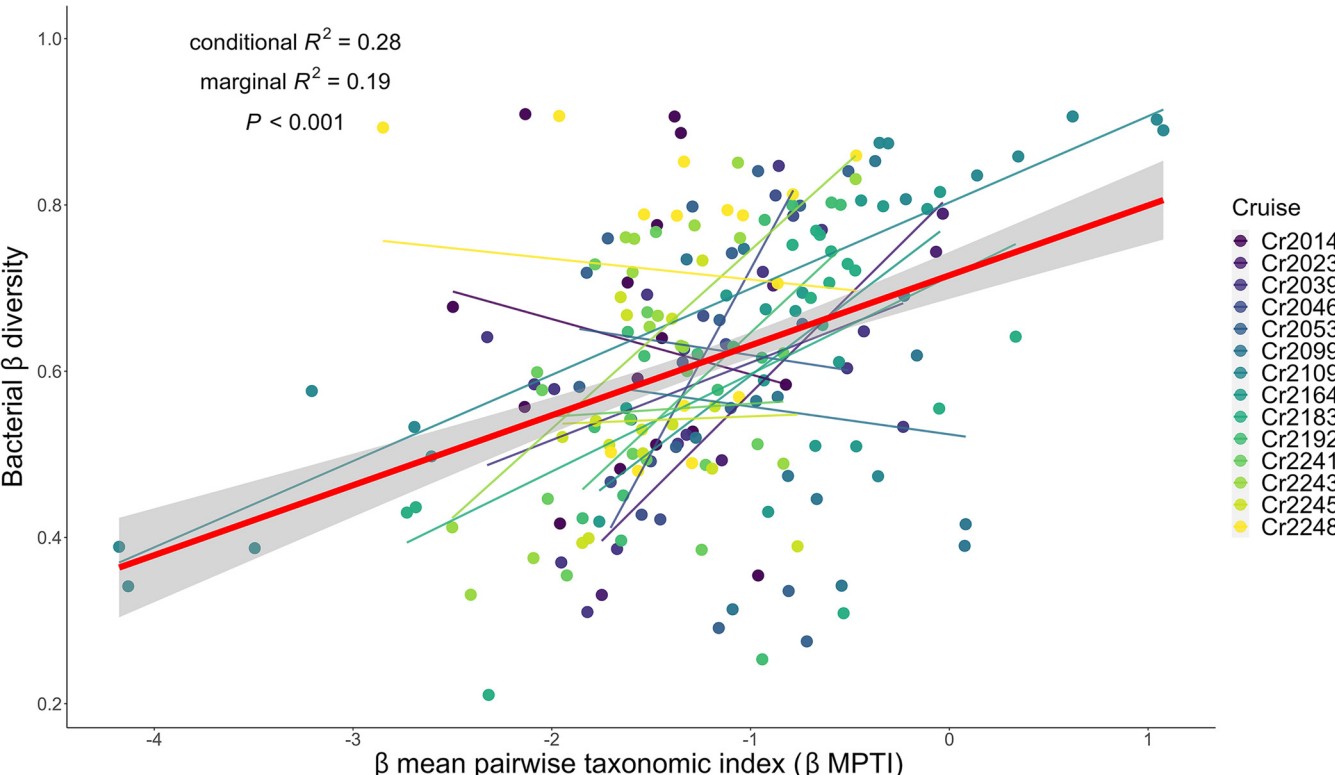

**FIG 3** Association between bacterial $\beta$ diversity and community assembly processes ($\beta$MPTI). The bold red line indicates the significant regression line fitted by the GLMM, with cruise as the random effect. The shaded area represents the confidence interval of regression. The colored dots represent measurements from different sampling cruises. The thin lines are the regression lines fitted by the GLM for each sampling cruise.

variable $\beta$ diversity in the literature could stem from the difference among ecosystem types, where different levels of habitat heterogeneity, dispersal limitation, and abiotic influences will alter the relationship between biodiversity and ecosystem functioning (1). However, because we did not compare $\beta$ diversity across ecosystems, the systematic dis-

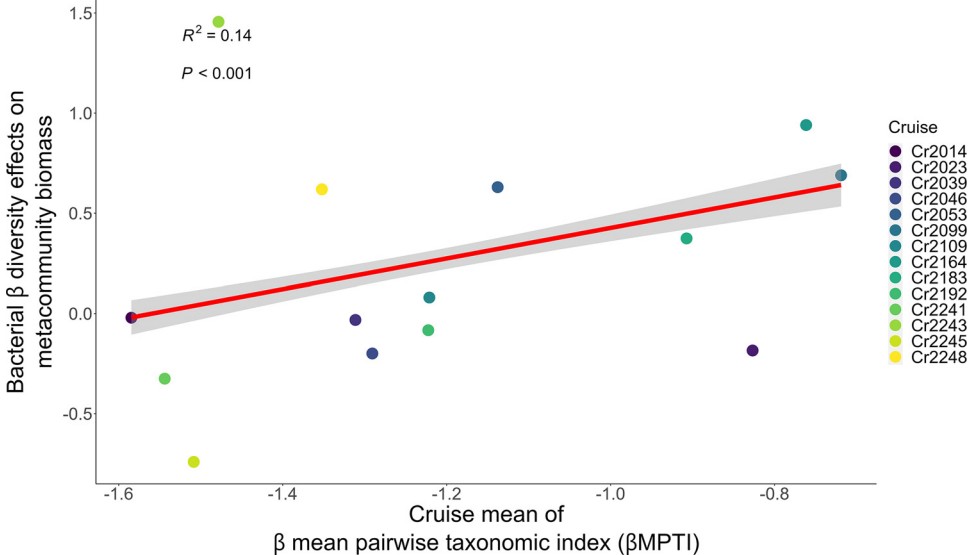

**FIG 4** Positive association between bacterial $\beta$ diversity effects on the sum of bacterial community biomass and community assembly processes ($\beta$MPTI) across cruises. The red line indicates the significant regression line fitted by the GLM. The shaded area represents the confidence interval of regression. Each of the colored dots represents the slope extracted from the GLM regressing the sum of bacterial community biomass on bacterial $\beta$ diversity for one of the 14 sampling cruises (cf. Fig. 2).

**TABLE 2** Results of GLM testing hypothesis II[a]

| Independent variable[b] | Regression coefficient | SE | P |
|---|---|---|---|
| $\beta$MPTI | 0.79 | 0.06 | <0.01 |
| Log (salinity) | −22.81 | 1.53 | <0.01 |
| Log (total inorganic nitrogen) | 0.4 | 0.02 | <0.01 |
| Log (phosphate) | −0.63 | 0.03 | <0.01 |
| Log (PAR) | −0.05 | 0.02 | <0.01 |
| Log (chlorophyll $a$) | 0.37 | 0.03 | <0.01 |

[a]Hypothesis II is that based on cross-cruise comparison, the bacterial $\beta$ diversity effect is stronger (more positive) when $\beta$ diversity in increased more by deterministic and less by stochastic assembly processes (Fig. 1C). The model presented here is the results after backward stepwise selection to statistically control for potential confounding factors. Steps of the backward stepwise selection are listed in Table S3. The results indicate that, after accounting for environmental variables, the effect of bacterial $\beta$ diversity on the summed bacterial community biomass increased with deterministic processes selecting for dissimilar species. This conclusion is qualitatively the same as that drawn from the univariate GLMM (Fig. 3).
[b]Bacterial $\beta$ diversity effects on the summed bacterial community biomass was the dependent variable.

crepancy among ecosystem types should not be responsible for the variable diversity effect in our study. Instead, we found the variability of $\beta$ diversity effects coincided with the variability of community assembly processes, as indicated by $\beta$MPTI (Fig. 3 and Fig. S3). Accordingly, we argue that the variable $\beta$ diversity effects may be caused by different strengths of deterministic community assembly processes influencing the $\beta$ diversity (8).

Through cross-cruise analysis, we showed that $\beta$ diversity effects became more positive when $\beta$ diversity was increased by deterministic assembly processes selecting for phylogenetically dissimilar species (Fig. 4). This finding is analogous to the studies revealing that bacteria with certain metabolic capabilities are selected by deterministic assembly processes, so that the bacterial communities can utilize resources in the environment; these selected bacterial communities in turn determine the ecosystem functioning (51–53). Also relevant but to a lesser extent, other studies show that the function of multiple communities is enhanced when deterministic assembly processes select for species that are dissimilar from each other (54, 55). Based on our data together with these studies, we argue that when bacterial communities are governed more by deterministic, and less by stochastic, assembly processes, bacterial species may experience some niche-based processes, such as interspecific interactions or interactions with the environments (20, 26, but see 27). Bacterial species that undergo some biotic and abiotic interactions are more adapted to survive in an environment (9), so that they should be more capable of capturing resources and produce more biomass. However, we ought to make such arguments with caution, because deterministic processes do not necessarily map to niche-based processes (27, 56, 57). Other specific processes potentially contributing to the influences of deterministic versus stochastic processes, such as dispersal (51, 54), cannot be clearly deciphered in our analyses. Our argument here remains to be validated by further studies that explicitly examine specific niche-based assembly processes and their influences on the biodiversity-ecosystem functioning relationship.

When investigating the influence of certain community assembly processes, previous research has been dedicated to study dispersal, and these studies have shown that diversity is maximized at intermediate levels of dispersal that reduce local environmental and demographic stochasticity without imposing too-strong homogenization (58–60). However, whether our study is comparable to this line of research is debatable, because the influence of dispersal was not explicitly quantified in our analyses, and we are not able to pinpoint dispersal on the deterministic-stochastic spectrum. Moreover, whether dispersal is a more deterministic or stochastic assembly process might depend on species and cannot be unarguably defined (61, 62). Since dispersal cannot be easily defined and parsed out, we refrain from making inferences based on dispersal or other specific processes like species interactions. In fact, this is one of the limitations of our analytical framework. Our analyses should be viewed as an examination of collective influences of deterministic versus stochastic assembly processes but not an examination of specific ecological processes. Future studies

dedicated to examining the influences of specific assembly processes are required and encouraged to validate our arguments.

Some caveats associated with our analytical framework are worth mentioning. First, we are aware that interdependency among $\alpha$, $\beta$, and $\gamma$ diversity could lead to statistical artifacts (11, 13). We did find that $\beta$ diversity was correlated with the richness and Shannon diversity of the bacterial communities (correlation coefficients = 0.14 and 0.18, respectively; both $P$ values < 0.01) (Fig. S4A and B). The richness and Shannon diversity were calculated from the combined communities of any pair of the 6 sampling sites in the same sampling cruise. However, the summed bacterial community biomass was not significantly associated with the richness and Shannon diversity (Fig. S4C and D), so the effect of $\beta$ diversity should not be caused by the confounding correlation between $\alpha$ and $\beta$ diversity. Moreover, including bacterial $\alpha$ diversity as a covariate for testing the first hypothesis did not negate the positive effect of bacterial $\beta$ diversity on the summed community biomass (Fig. S5). We thus argue that the positive effect of bacterial $\beta$ diversity is robust to the interdependency between $\alpha$ and $\beta$ diversity.

We are also aware that the community biomass is not a perfect representation of ecosystem functioning; however, it has been commonly used (63–67). Biomass *per se* does not necessarily reflect the resource capture, nutrient recycling, or decomposition rate of a community (1). The functioning of bacterial community should be quantified as a kind of metabolic rate, like enzyme activities (55). Unfortunately, measurements of bacterial metabolism are not available in our data set. Future studies applying metagenomic or metatranscriptomic analyses to our analytical framework will constitute a direct test of the biodiversity-ecosystem functioning relationship.

Next, our findings hinge on the assumption that $\beta$MPTI, or phylogenetic similarity in general, can be used to infer the collective influences of community assembly processes. First, we acknowledge that phylogenetic similarity might not faithfully approximate community assembly processes (68, 69). However, we observed significant phylogenetic signals (Fig. S6), which somehow demonstrated the relationship between phylogenetic relatedness and ecological similarity (34). Second, we calculated $\beta$MPTI in a fashion similar to that used for the net related index (NTI), nearest taxon index (NRI) (33, 70), $\beta$NTI, and $\beta$NRI (71). $\beta$NTI and $\beta$NRI were been used to infer the influences of assembly processes that determine the composition of microbial community (72–74). We thus cautiously treat the $\beta$MPTI as an implication of collective influences of community assembly processes along the deterministic-stochastic spectrum. When the collective influence is more deterministic, we then argue with caution that the communities are subject to certain processes, e.g., environmental filtering (75) or species interaction (76).

Despite these limitations, our analytical framework expands the current biodiversity-ecosystem functioning framework from single communities to multiple communities. Because an ecological community almost never exists alone in nature, expanding to multiple communities allows us to holistically perceive the role of biodiversity in natural ecosystems. In addition, we demonstrate how the influence of community assembly processes could also alter the effect of $\beta$ diversity on the overall function of multiple communities. Taking the influences of community assembly processes into account equips us with a more mechanistic understanding of the impacts of biodiversity on ecosystem functioning.

In conclusion, we showed that bacterial $\beta$ diversity positively affected the summed biomass of bacterial communities in the southern ECS. Moreover, this positive effect was stronger when bacterial $\beta$ diversity was increased by the assembly processes that deterministically (i.e., nonrandomly) selected for phylogenetically dissimilar species. We infer from our findings that when some community assembly processes deterministically influence the bacterial communities in the southern ECS, the bacterial $\beta$ diversity has a more positive effect on the overall functions of these communities.

## MATERIALS AND METHODS

**Bacterial community composition and environmental variables.** We collected bacteria from a transect of 6 sampling sites in the southern ECS (see the map in Fig. 1A). We visited this transect in

14 cruises from April 2014 to July 2017, so that a total of 84 samples were obtained. At each sampling site, GoFlo bottles (General Oceanics) mounted on a conductivity, temperature, and depth profiler (CTD profiler; Sea-Bird Electronics, Bellevue, WA, USA) were used to collect ~20 L of seawater from 5 m beneath the surface. The seawater was prefiltered through a screen mesh with 20-$\mu$m openings to remove large particles like zooplankton. With the prefiltered seawater, we revealed the taxonomic composition with high-throughput sequencing techniques.

The prefiltered seawater was filtered sequentially through a 1.2-$\mu$m- and a 0.2 $\mu$m-pore-size filter (Millipore Isopore hydrophilic polycarbonate membrane). The 0.2-$\mu$m-pore-size filter was frozen immediately with liquid nitrogen and then stored at −80°C before molecular analysis (77). From the filter, we extracted and sequenced DNA and processed sequences for bacterial composition. The methods of DNA extraction and sequence processing are described briefly here; detailed methods are presented in Text S1. Total DNA was extracted from the 0.2 $\mu$m-pore size filter with the PowerWater DNA extraction kit (Qiagen) according to the manufacturer's instructions. DNA extracts from the filters were used as templates for PCR to amplify the 16S rRNA gene for bacterial community analysis. PCR was performed in two steps to gain better reproducibility and consistent results (78); see Text S1 for details of the two-step PCR procedures. After obtaining 16S rRNA gene sequences, the DADA2 pipeline was used for quality filtering and assembling sequences into amplicon sequence variants (ASVs) (79); see Text S1 for sequence merging procedures. Taxonomy assignment was performed on ASVs to recognize and select for those classified under the bacteria kingdom based on the Silva 132 database (80). Finally, in order to obtain the phylogeny, we used a maximum-likelihood method (with a negative-edge length of 0) (79) to build phylogenetic trees for bacteria from 16S rRNA genes from all cruises.

For each sampling station, environmental variables, including temperature, salinity, and photosynthetic active radiation, were recorded by the CTD profiler. In addition, 100-mL seawater samples for chlorophyll $a$ and total dissolved inorganic nitrogen and phosphate concentrations were collected and measured according to standard methods (42).

**Bacterial community biomass and $\beta$ diversity.** To estimate the bacterial community biomass of each sampling site, we preserved 10 mL of prefiltered seawater in glutaraldehyde with a final concentration of 0.1%. The samples were frozen with liquid nitrogen and stored at −80°C before counting with a flow cytometer. Prior to processing with the flow cytometer, the samples were stained with SYBR green I and then incubated for 15 min at room temperature in the dark. We determined bacterial density with the CyFlow Space cytometer (Partec) at a rate allowing <1,000 events per s to avoid particle coincidence. Bacterial density was then used to estimate bacterial community biomass with a carbon conversion factor of $2 \times 10^{-14}$ g C/cell (81). To account for the potential predators of bacteria, we counted the heterotrophic nanoflagellates (HNF) under a fluorescence microscope (82). Specifically, we additionally preserved 50 mL of prefiltered seawater in neutralized formalin with a final concentration of 2% and stored the samples at 4°C. Samples were identified and counted, using an inverted epifluorescence microscope (Nikon TMD 300) at ×200 or ×400 magnification (83). Finally, HNF community biomass was converted using a carbon conversion factor of $4.7 \times 10^{-12}$ g C/cell (84).

For any pair of the 6 sampling sites (15 pairs in total) within a cruise, we calculated the Bray-Curtis dissimilarity index and the summed bacterial community biomass to represent the bacterial $\beta$ diversity and as a proxy for the overall function of the paired communities, respectively. Before calculating the $\beta$ diversity of bacteria, each community was resampled once to achieve the same number of reads, i.e., 13,129, the minimum reads among stations across cruises. Although this procedure addressed the disparity issue (i.e., unequal reads among stations), it is still not appropriate to compare the relative abundance of AVSs across stations (85). Therefore, we applied the method of Chao et al. to rarefy bacterial communities in order to have a fair among-station comparison (86–88). With the rarefied compositions, we calculated the Bray-Curtis dissimilarity for any pair of the 6 sampling sites as the $\beta$ diversity.

**Community assembly processes.** To infer the community assembly processes, we calculated the $\beta$ mean pairwise taxonomic index ($\beta$MPTI). The $\beta$MPTI extends Webb's net relatedness index (33, 70) from within a community across two communities (71). The $\beta$MPTI quantifies the deviation of the observed mean pairwise phylogenetic distance (MPD$_{obs}$) in two communities from a null distribution of mean pairwise phylogenetic distance (MPD) in the same two communities. The MPD$_{obs}$ was calculated as the mean branch length among all pairs of species in the two communities. The null distribution of the MPD is generated by random processes so that the distribution represents the values of the MPD if the two communities are assembled completely by random processes. The deviation of MPD$_{obs}$ from the null thus represents the nonrandom/deterministic processes by which the two communities are assembled. To generate a null distribution of MPD by random processes, we first randomly shuffled the phylogeny (i.e., randomly shuffled the tips of the phylogenetic tree) of all species in two communities (i.e., any pairs of the 6 stations within a cruise). With the randomized phylogeny, we calculated the MPD of the two communities. Repeating this randomization technique 999 times, we generated the null distribution of MPD. Finally, $\beta$MPTI was calculated as (MPD$_{obs}$ − mean MPD$_{null}$)/SD MPD$_{null}$, where mean MPD$_{null}$ represents the mean of the null distribution of MPD and SD MPD$_{null}$ represents the standard deviation of the null distribution of MPD.

The $\beta$MPTI is nearly identical to the $\beta$ net relatedness index ($\beta$NTI). The only difference is that we multiply $\beta$NTI by −1, so that the sign of $\beta$MPTI intuitively represents whether the observed phylogenetic distance is more similar (more negative) or less similar (less negative) to random. A negative $\beta$MPTI means that species in the two communities are more phylogenetically similar than expected from random. A more negative $\beta$MPTI thus implies that the two communities are subject to assembly processes selecting for species that are phylogenetically more similar to each other than expected from

random. Following the same logic, a less negative or positive $\beta$MPTI hints at the assembly processes that select for phylogenetically less similar (i.e., more dissimilar) species than expected from random.

**Biodiversity-ecosystem functioning relationships and impacts of community assembly processes.** With the bacterial $\beta$ diversity and biomass, we tested hypothesis I (that bacterial $\beta$ diversity increases the summed bacterial community biomass) (Fig. 1B). To do so, we used a GLMM to regress the sum of bacterial community biomass against $\beta$ diversity, with cruise as the random effect. The sum of bacterial community biomass and $\beta$ diversity was log transformed to improve normality. Making cruise a random effect should account for within-cruise autocorrelations. To statistically account for potential confounding factors, we conducted backward selection by first including all environmental variables and removing the variables that were not significant based on $P$ values in a stepwise fashion. The environmental variables include the average temperature, salinity, total dissolved inorganic nitrogen, phosphate, photosynthetically active radiation (PAR), and chlorophyll $a$ concentration (collected at the same depth as the bacterial samples) of the two sampling sites.

Next, we proceeded to test hypothesis II (that based on cross-cruise comparison, the bacterial $\beta$ diversity effect is stronger [more positive] when $\beta$ diversity is increased more by deterministic and less by stochastic assembly processes) (Fig. 1C). First, we regressed the bacterial $\beta$ diversity against $\beta$MPTI, in order to check whether, in general, $\beta$ diversity increased with assembly processes that selected for phylogenetically dissimilar species. Next, the effect of $\beta$ diversity was estimated for each cruise by conducting GLM that regressed the summed bacterial community biomass on $\beta$ diversity of any pair of the 6 sampling sites within a cruise. Here, the resulting 14 regression coefficients (i.e., slopes) were regarded as the effects of $\beta$ diversity on the overall function of the two communities. We finally regressed these regression coefficients against the mean $\beta$MPTI of each sampling cruise, to test if the effects of $\beta$ diversity would increase with the assembly processes that selected for phylogenetically dissimilar species. Similarly, we also conducted backward selection to statistically account for potential confounding environmental variables.

**Computation.** We used the phyloseq package to perform sequence subsampling to achieve parity in total number of reads (89), the iNEXT package to perform rarefaction (88), the phangorn package to build phylogenetic trees (79), the picante package to calculate phylogenetic distances to derive $\beta$MPTI (90), and the nlme package to perform generalized linear mixed-effect models (GLMM) (91). All packages were built and computation was carried out in R ver. 4.1.1 (92).

**Data availability.** The sequence data have been deposited in the NCBI Sequence Read Archive (SRA) under the accession number PRJNA662424.

## SUPPLEMENTAL MATERIAL

Supplemental material is available online only.

**TEXT S1**, DOCX file, 0.02 MB.
**FIG S1**, TIF file, 4.7 MB.
**FIG S2**, TIF file, 2.5 MB.
**FIG S3**, TIF file, 8.2 MB.
**FIG S4**, TIF file, 0.5 MB.
**FIG S5**, TIF file, 0.3 MB.
**FIG S6**, TIF file, 3.3 MB.
**TABLE S1**, DOCX file, 0.03 MB.
**TABLE S2**, DOCX file, 0.03 MB.
**TABLE S3**, DOCX file, 0.02 MB.

## ACKNOWLEDGMENTS

We thank Hon-Tsen Yu for providing facilities and advice on laboratory work. We also thank Sara Jackrel and Wenxue Wu for feedback on the manuscript.

This work was supported by the National Center for Theoretical Sciences, Foundation for the Advancement of Outstanding Scholarship, and the National Science and Technology Council, Taiwan.

We declare no competing financial interests.

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
