## [Reviewer comments · mSystems]

Deterministic assembly processes strengthen the effects of β -diversity on community biomass of marine bacterioplankton

Feng-Hsun Chang, Jinny Yang, Ariana Chih-Hsien Liu, Hsiao-Pei Lu, Gwo-Ching Gong, Fuh-Kwo Shiah, and Chih-hao Hsieh

Corresponding Author(s): Feng-Hsun Chang, National Taiwan University

Review Timeline:

Submission Date:

October 6, 2022

Accepted:

November 21, 2022

Editor: Michaeline Albright

Reviewer(s): The reviewers have opted to remain anonymous.

Transaction Report:

DOI: <https://doi.org/10.1128/msystems.00970-22>

November 21, 2022

Dr. Feng-Hsun Chang
National Taiwan University
Institute of Oceanography
Taipei City 10617
Taiwan

Re: mSystems00970-22 (Deterministic assembly processes strengthen the effects of β -diversity on community biomass of marine bacterioplankton)

Dear Dr. Feng-Hsun Chang:

Thank you for thoroughly responding to the comments raised during the previous review of the manuscript.

Your manuscript has been accepted, and I am forwarding it to the ASM Journals Department for publication. For your reference, ASM Journals' address is given below. Before it can be scheduled for publication, your manuscript will be checked by the mSystems production staff to make sure that all elements meet the technical requirements for publication. They will contact you if anything needs to be revised before copyediting and production can begin. Otherwise, you will be notified when your proofs are ready to be viewed.

Publication Fees:

If you would like to submit a potential Featured Image, please email a file and a short legend to mSystems@asmusa.org. Please note that we can only consider images that (i) the authors created or own and (ii) have not been previously published. By submitting, you agree that the image can be used under the same terms as the published article. File requirements: square dimensions (4" x 4"), 300 dpi resolution, RGB colorspace, TIF file format.

We recognize that the video files can become quite large, and so to avoid quality loss ASM suggests sending the video file via <https://www.wetransfer.com/>. When you have a final version of the video and the still ready to share, please send it to mSystems staff at mSystems@asmusa.org.
